# Towards Adversarially Robust Condensed Dataset by Curvature Regularization

## Abstract

Dataset condensation is a recent technique designed to mitigate the rising computational demands of training deep neural networks. It does so by generating a significantly smaller, synthetic dataset derived from a larger one. While an abundance of research has aimed at improving the accuracy of models trained on synthetic datasets and enhancing the efficiency of synthesizing these datasets, there has been a noticeable gap in research focusing on analyzing and enhancing the robustness of these datasets against adversarial attacks. This is surprising considering the appealing hypothesis that condensed datasets might inherently promote models that are robust to adversarial attacks. In this study, we first challenge this intuitive assumption by empirically demonstrating that dataset condensation methods are not inherently robust. This empirical evidence propels us to explore methods aimed at enhancing the adversarial robustness of condensed datasets. Our investigation is underpinned by the hypothesis that the observed lack of robustness originates from the high curvature of the loss landscape in the input space. Based on our theoretical analysis, we propose a new method that aims to enhance robustness by incorporating curvature regularization into the condensation process. Our empirical study suggests that the new method is capable of generating robust synthetic datasets that can withstand various adversarial attacks.

## 1 Introduction

In the era of big data, the computational demands for training deep learning models are continuously growing due to the increasing volume of data. This presents substantial challenges, particularly for entities with limited computational resources. To mitigate such issues, concepts like dataset distillation (Wang et al., 2018) and dataset condensation (Zhao et al., 2021; Zhao & Bilen, 2021; 2023) have emerged, offering a means to reduce the size of the data while maintaining its utility.

Dataset condensation specifically refers to the task of synthesizing a smaller dataset such that models trained on this smaller set yield high performance when tested against the original, larger dataset. The dataset condensation algorithm takes a large dataset as input and generates a compact, synthetic dataset. The efficacy of this condensed dataset is evaluated by training models on it and subsequently testing these models on a separate, real dataset.

The successful implementation of dataset condensation can bring many benefits, such as enabling more cost-effective research on large datasets. Consequently, recent research has expanded rapidly in this direction, with numerous studies examining different aspects of the process, with most research efforts focusing on either improving the accuracy of models trained on condensed datasets or enhancing the efficiency of the condensation procedure. Less attention, however, has been given to other crucial properties such as the adversarial robustness of these datasets. Adversarial robustness is integral to the condensation process, particularly considering the fact that dataset condensation holds much potential in the future landscape of trustworthy machine learning, as highlighted by recent surveys (Geng et al., 2023; Chen et al., 2023).

One intriguing area for exploration is the possibility that adversarial robustness might be inherently induced by the dataset condensation process. The hypothesis that underpins our study is that during the condensation process, the algorithms aim to condense larger datasets while maintaining the level of accuracy associated with these larger sets. A plausible approach to achieve this is to preserve the salient patterns in the images as much as possible and discard the non-essential information

embedded in the original images. This relationship presents an intriguing link to studies focusing on adversarial robustness, as previous work suggests that the adversarial vulnerability of models is often tied to these non-observable features (Ilyas et al., 2019; Wang et al., 2020).

Drawing upon this connection, our research is driven by a fundamental question:

*Do dataset condensation algorithms inherently generate datasets that can foster adversarially robust models?*

while our initial experiments quickly suggest a negative response, we continued to explore a more challenging question:

*How can we incorporate adversarial robustness into the dataset condensation process, thereby generating datasets that lead to more robust models?*

Motivated by this inquiry, we explored the theoretical link between adversarial robustness and the curvature of the loss function. We propose a novel method, GUARD (Geometric regUlarization for Adversarial Robust Dataset) , which incorporates curvature regularization into the condensation process. By aligning the principles of gradient matching, we aim to create a synthetic dataset that synchronizes the gradients of the model trained on this synthetic dataset with those of a model trained under robust conditions. Our proposed method is comprehensively evaluated against existing condensation methods on MNIST and CIFAR10 datasets.

In summary, our contributions of this paper are as follows

- Empirical and theoretical exploration of adversarial robustness in condensed synthetic datasets
- Introduction of a new theory-motivated method, GUARD, that offers robust dataset condensation.
- Presentation of the first comprehensive adversarial robustness benchmark for existing state-of-the-art dataset condensation methods.

The remainder of this paper is structured as follows. In Section 2, we introduce a range of related works that provide context for our research. In Section 3, we offer a concise overview of dataset condensation and delve into the specific formulations of the two dataset condensation methods pertinent to our approach. Section 4 examines the unexpected lack of robustness with dataset condensation, despite intuitive reasoning that suggests otherwise. Section 5 presents a theoretical bound on robustness, which we employ to introduce our method, GUARD, in Section 6. Our findings are subsequently detailed in Section 7, and we conclude the paper with a summary in Section 8.

## 2 RELATED WORKS

**Datasets Condensation**    Dataset condensation is a technique that has been developed to address the issue of the increasing amount of data required to train deep learning models. The goal of dataset condensation is to efficiently train neural networks using a small set of synthesized training examples from a larger dataset. Dataset distillation (DD) (Wang et al., 2018) was one of the first such methods developed, and it showed that training on a few synthetic images can achieve similar performance on MNIST and CIFAR10 as training on the original dataset. Later, Cazenavette et al. (2022); Zhao & Bilen (2021); Zhao et al. (2021) explored different methods of condensation, including gradient and trajectory matching. These approaches focused on matching the gradient w.r.t. the real and synthetic data, with stronger supervision for the training process. Instead of matching the weights of the neural network, another thread of works (Lee et al., 2022; Wang et al., 2022) focuses on matching feature distributions of the real and synthetic data in the embedding space to better align features or preserve real-feature distribution. Considering the lack of efficiency of the bilevel optimization in previous methods, Nguyen et al. (2021); Zhou et al. (2022) aim to address the significant amount of meta gradient computation challenges. Nguyen et al. (2020) proposed a kernel-inducing points meta-learning algorithm and they further leverage the connection between the infinitely wide ConvNet and kernel ridge regression for better performance. Furthermore, Sucholutsky & Schonlau (2021) focuses on simultaneously distilling images and the paired soft labels. These approaches can be

broadly classified into four families based on their underlying principles: meta-model matching, gradient matching, distribution matching, and trajectory matching (Sachdeva & McAuley, 2023).

**Adversarial Attacks**    Adversarial attacks are a significant concern in the field of machine learning, as they can cause models to make incorrect predictions even when presented with seemingly similar input. Many different types of adversarial attacks have been proposed in the literature (Goodfellow et al., 2015; Madry et al., 2018). In particular, Projected Gradient Descent (PGD) is a widely used adversarial attack that has been shown to be highly effective against a variety of machine learning models (Madry et al., 2018). Kurakin et al. (2017) demonstrates the real-world implications of these attacks. The limitations of defensive distillation, a technique initially proposed for increasing the robustness of machine learning models, were explored by Papernot et al. (2016b). Moosavi-Dezfooli et al. (2016) introduced DeepFool, an efficient method to compute adversarial perturbations. In a similar vein, Carlini & Wagner (2017) developed a procedure to increase the robustness of an arbitrary neural network. Other notable works include the introduction of transferability in adversarial attacks byPapernot et al. (2016a), the simple and effective black-box attack by Narodytska & Kasiviswanathan (2016), and the zeroth-order optimization-based attack by Chen et al. (2017). More recently, Athalye et al. (2018) investigated the robustness of obfuscated gradients, and Wong et al. (2019) introduced the Wasserstein smoothing as a novel defense against adversarial attacks. Croce & Hein (2020) introduced AutoAttack, which is a more recent suite of adversarial attacks. It consists of four diverse and parameter-free attacks that are designed to provide a comprehensive evaluation of a model's robustness to adversarial attacks.

**Adversarial Defense**    Numerous defenses against adversarial attacks have been proposed. Among these, adversarial training stands out as a widely adopted defense mechanism that entails training machine learning models on both clean and adversarial examples (Goodfellow et al., 2015). Several derivatives of the adversarial training approach have been proposed, such as ensemble adversarial training (Tramèr et al., 2018), and randomized smoothing (Cohen et al., 2019) — a method that incorporates random noise to obstruct the generation of effective adversarial examples. However, while adversarial training can be effective, it bears the drawback of being computationally expensive and time-consuming.

Some defense mechanisms adopt a geometrical approach to robustness. One such defense mechanism is Curvature Regularization (CURE), a method that seeks to improve model robustness by modifying the loss function used during training (Moosavi-Dezfooli et al., 2019). The primary aim of curvature regularization is to minimize the sensitivity of the model, to adversarial perturbations in the input space and it is more difficult for an attacker to find adversarial examples which cross this boundary. Miyato et al. (2015) focuses on improving the smoothness of the output distribution to make models more resistant to adversarial attacks, while Cisse et al. (2017b) introduced Parseval networks, which enforce Lipschitz constant to improve model robustness. Ross & Doshi-Velez (2018) presented a method for improving the robustness of deep learning models using input gradient regularization.

Several other types of defense techniques have also been proposed, such as corrupting with additional noise and pre-processing with denoising autoencoders by Gu & Rigazio (2014), the defensive distillation approach by Papernot et al. (2016c), and the Houdini adversarial examples by Cisse et al. (2017a).

## 3    PRIMER: DATASET CONDENSATION

Before we delve deeper into the theory of robustness in dataset condensation methods, we will formally introduce their formulation in this section. Although there are different families of dataset condensation methods, the central objective remains the same. The goal is to learn a synthetic dataset $\mathcal{S}$ with $|\mathcal{S}|$ image-label pairs from a real dataset $\mathcal{T}$ with $|\mathcal{T}|$ image-label pairs, where $|\mathcal{S}|$ is much smaller than $|\mathcal{T}|$. Different methods accomplish this goal through different objective functions. In its most general form, we can formulate the dataset condensation problem as

$$\min_{\mathcal{S}} \boldsymbol{F}(\mathcal{S}, \mathcal{T})$$

where $\boldsymbol{F}$ is some objective function that differs by each method. The original dataset condensation method (DC) (Zhao et al., 2021) seeks to enable some network trained on the synthetic dataset to converge to a similar parameter compared to training on the real dataset. Hence, it is formulated as the problem

$$\min_{\mathcal{S}} D(\boldsymbol{\theta}^{\mathcal{S}}, \boldsymbol{\theta}^{\mathcal{T}}) \text{ where } \boldsymbol{\theta}^{\mathcal{S}}(\mathcal{S}) = \arg\min_{\boldsymbol{\theta}} \mathcal{L}^{\mathcal{S}}(\boldsymbol{\theta})$$

where $\phi$ is some neural network and $D$ is a distance function in the parameter space (Zhao et al., 2021). To ensure the synthetic dataset works well on a range of network initializations, DC samples initial weights from a random distribution and is modified into the below form:

$$\min_{\mathcal{S}} E_{\boldsymbol{\theta}_0 \sim P_{\boldsymbol{\theta}_0}} D(\boldsymbol{\theta}^{\mathcal{S}}(\boldsymbol{\theta}_0), \boldsymbol{\theta}^{\mathcal{T}}(\boldsymbol{\theta}_0)) \text{ where } \boldsymbol{\theta}^{\mathcal{S}}(\mathcal{S}) = \arg\min_{\boldsymbol{\theta}} \mathcal{L}^{\mathcal{S}}(\boldsymbol{\theta}(\boldsymbol{\theta}_0))$$

It can be seen that this method is computationally expensive as it is a dual optimization problem that involves first optimizing $\boldsymbol{\theta}_{\mathcal{S}}$ with respect to the loss function $\mathcal{L}$, and then optimizing $\mathcal{S}$ with respect to the distance function to achieve a similar solution in the parameter space.

To address this issue, Dataset Condensation with Distribution Matching (DM) (Zhao & Bilen, 2023) was proposed. DM uses maximum mean discrepancy (MMD) to directly optimize the synthetic dataset, avoiding the need to train a network. Let $\psi$ be an embedding function with parameter $\boldsymbol{\theta}$ that embeds each image $\boldsymbol{x} \in \mathcal{R}^d$ into $\mathcal{R}^{d'}$, where $d' << d$. The DM problem can be formulated as

$$\min_{\mathcal{S}} E_{\boldsymbol{\theta} \sim P_{\boldsymbol{\theta}}} || \frac{1}{|\mathcal{T}|} \sum_{i=1}^{|\mathcal{T}|} \psi_{\boldsymbol{\theta}}(\boldsymbol{x}_i) - \frac{1}{|\mathcal{S}|} \sum_{j=1}^{|\mathcal{S}|} \psi_{\boldsymbol{\theta}}(\boldsymbol{s}_j) ||^2$$

This particular formulation will prove integral to subsequent proofs detailed later in the paper.

Regardless of the formulation, dataset condensation methods usually allow for a flexible condensation scale by setting $|\mathcal{S}|$ to different values when learning the synthetic dataset. However, instead of expressing the condensation scale as $|\mathcal{S}|$ directly, it is more common to express it in the unit of "images per class" (ipc).

## 4 DOES DATASET CONDENSATION CONTAIN ROBUSTNESS?

In this section, we delve into the robustness pertaining to dataset condensation methods. We are interested in this question because, from intuition, dataset condensation could eliminate confounding information and spurious features from images. Since dataset condensation usually needs to enable the model to learn some representations of each class using fewer than 50 images, any instance-specific features in the synthetic dataset would likely be replaced by the more generalizable intra-class features. Previous research suggests that these non-observable features can, at times, be exploited by adversarial attacks (Ilyas et al., 2019; Wang et al., 2020).

This hypothesis finds validation when we inspect the synthetic dataset generated by DC. In Figure 1, we present an example comprising two images from the CIFAR10 dataset, used as initial images in the synthetic dataset. Furthermore, we display the resulting image post-DC application. A noticeable observation is the there is limited color information encoded in the synthetic car image. This is due to cars often appearing in a wide color spectrum, and the image must facilitate model generalization across this spectrum. In contrast, the synthetic horse image distinctly encodes the red/brown color scheme typical of horses.

**Empirical results** To verify the robustness of dataset condensation methods, we evaluated the robustness of models trained on the synthetic data from a variety of methods, DD (Wang et al., 2018), DC (Zhao et al., 2021), MTT (Cazenavette et al., 2022), DiM (Wang et al., 2023), and FRePo (Zhou et al., 2022) under $\ell_{\infty}$ PGD attacks. The result is shown in Table 1. Contrary to our belief, the results showed that the resulting models have little robustness to adversarial attacks. PGD attack with default hyperparameters can diminish the accuracy to 0, and even attacks with much reduced

Table 1: Clean accuracy and robust accuracy on ConvNet trained on 10 images per class synthetic datasets produced by different methods, under $\ell_\infty$ PGD attacks. For MNIST, the parameters of the attack are $\epsilon = 0.3$, $\alpha = 0.1$, steps $= 10$. For CIFAR10, the parameters of the attack are $\epsilon = 8/255$, $\alpha = 2/255$, steps $= 10$.

|  |  | DD | DC | MTT | DiM | FRePo |
|---|---|---|---|---|---|---|
| MNIST | Clean | 79.5±8.1 | 96.4±0.1 | 97.3±0.1 | 98.6±0.2 | 98.6±0.1 |
|  | Robust | 0.0±0.0 | 0.0±0.0 | 0.0±0.0 | 0.0±0.0 | 0.0±0.0 |
| CIFAR10 | Clean | 36.8±1.2 | 45.3±0.5 | 65.3±0.7 | 66.2±0.5 | 65.5±0.4 |
|  | Robust | 1.0±0.3 | 0.3±0.1 | 0.1±0.0 | 0.0±0.0 | 0.9±0.6 |

intensity degrade the performance severely (see Table 2). While current dataset condensation methods effectively condense task-related information into a small set of parameters (usually dozens of times the number of pixels in an image), the resulting models are easily fooled by adversarial attacks.

## 5 THEORETICAL BOUND OF ROBUSTNESS

**Theoretical explanation** Previous work (Jetley et al., 2018; Fawzi et al., 2018) has studied the adversarial robustness of neural networks via the geometry of the loss landscape. Here we find connections between standard training and dataset condensation to provide a theoretical explanation on the observed adversarial vulnerability of models trained with the standard dataset condensation methods.

Let $\ell(\mathbf{x}, y; \theta)$ denote the loss function of the neural network, or $\ell(\mathbf{x})$ for simplicity, and $\mathbf{v}$ is a perturbation vector. By Taylor's Theorem,

$$\ell(\mathbf{x} + \mathbf{v}) = \ell(\mathbf{x}) + \nabla\ell(\mathbf{x})^\top \mathbf{v} + \frac{1}{2}\mathbf{v}^\top \mathbf{H}\mathbf{v} + o(\|\mathbf{v}\|^2) \quad (1)$$

We are interested in the property of $\ell(\cdot)$ in the locality of $x$, so we focus on the quadratic approximation $\tilde{\ell}(\mathbf{x} + \mathbf{v}) = \ell(\mathbf{x}) + \nabla\ell(x)^\top \mathbf{v} + \frac{1}{2}\mathbf{v}^\top \mathbf{H}\mathbf{v}$. Define the adversarial loss on real data as $\tilde{\ell}_\rho^{adv}(\mathbf{x}) = \max_{\|\mathbf{v}\| \leq \rho} \tilde{\ell}(\mathbf{x} + \mathbf{v})$, we can expand this and take the expectation over the distribution with class label $c$, denoted as $D_c$

$$\mathop{\mathbb{E}}_{\mathbf{x} \sim D_c} \tilde{\ell}_\rho^{adv}(\mathbf{x}) \leq \mathop{\mathbb{E}}_{\mathbf{x} \sim D_c} \ell(\mathbf{x}) + \rho \mathop{\mathbb{E}}_{\mathbf{x} \sim D_c} \|\nabla\ell(\mathbf{x})\| + \frac{1}{2}\rho^2 \mathop{\mathbb{E}}_{\mathbf{x} \sim D_c} \lambda_1(\mathbf{x}) \quad (2)$$

where $\lambda_1$ is the largest eigenvalue of the Hessian matrix $\mathbf{H}(\ell(\mathbf{x}))$.

Then, we have the proposition:

Figure 1: A comparative analysis of images from CIFAR10 and the synthetic dataset generated using DC. The left column showcases original images from CIFAR10, while the right column exhibits the corresponding images post 1000 iterations of DC under the 1 ipc setting. The top pair of images represent a car, and the lower pair represent a horse.

**Proposition 1** *Let $\mathbf{x}'$ be a datum distilled from the training samples with the label $c$, and satisfies $\|h(\mathbf{x}') - \mathbb{E}_{\mathbf{x} \sim D_c}[h(\mathbf{x})]\| \leq \sigma$. Assume $\ell(\cdot)$ is convex in $\mathbf{x}$ and $\tilde{\ell}_\rho^{adv}(\cdot)$ is L-Lipschitz in the feature space, then the below inequality holds*

$$\tilde{\ell}_\rho^{adv}(\mathbf{x}') \leq \mathop{\mathbb{E}}_{\mathbf{x} \sim D_c} \ell(\mathbf{x}) + \rho \mathop{\mathbb{E}}_{\mathbf{x} \sim D_c} \|\nabla\ell(\mathbf{x})\| + \frac{1}{2}\rho^2 \mathop{\mathbb{E}}_{\mathbf{x} \sim D_c} \lambda_1(\mathbf{x}) + L\sigma \quad (3)$$

Because $\ell(\mathbf{x})$ and $\sigma$ are typically small, the above inequality shows that the upper bound of $\tilde{\ell}_\rho^{adv}(\mathbf{x}')$ is largely affected by the smoothness of the loss function in the locality of real data samples. In Appendix A, we give a more thorough proof of the proposition and discuss the validity of some of the assumptions made.

We wish to highlight a fundamental challenge associated with robust dataset condensation, which is that there is always a distribution shift between the real and condensed datasets. This shift

raises uncertainties about whether the enhanced robustness observed in the condensed dataset will be effectively transferred when evaluated against the real dataset. Nevertheless, our theoretical framework offers assurances regarding this concern. A comparison between Eq. 2 with Eq. 3 reveals that the bounds of adversarial loss for real data and a distilled datum differ only by $L\sigma$. We have thus demonstrated that the disparity between minimizing adversarial loss on the condensed dataset and doing so on the real dataset is confined to this constant. This finding allows for robust dataset condensation methods to exclusively enhance robustness with respect to the condensed dataset.

# 6 METHODS

## 6.1 GEOMETRIC REGULARIZATION FOR ADVERSARIAL ROBUST DATASET

Let $\mathcal{T} = \{\mathbf{x}_i, y_i\}_{i=1}^n$ denote the real dataset and $\mathcal{S} = \{\mathbf{x}_i', y_i'\}_{i=1}^m$ denote the synthetic dataset. Robust dataset condensation can be formulated as a tri-level optimization problem as below:

$$\min_{\mathcal{S}} \sum_{i=1}^n \max_{\|\mathbf{v}\| \leq \rho} \ell((\mathbf{x}_i + \mathbf{v}), y_i; \boldsymbol{\theta}(\mathcal{S})), \tag{4}$$

$$\text{where} \ \ \boldsymbol{\theta}(\mathcal{S}) = \arg\min_{\boldsymbol{\theta}} \sum_{i=1}^m \ell(\mathbf{x}_i', y_i') \tag{5}$$

Note that $\max_{\|\mathbf{v}\| \leq \rho} \ell((\mathbf{x}_i + \mathbf{v}), y_i; \theta)$ corresponds to the $\ell_\rho^{adv}(\mathbf{x})$ we have discussed above, which is different from the standard loss $\ell(\mathbf{x})$ used in most of the current dataset condensation methods. While recent theoretical works have led to different arguments as to whether accuracy is in principle at odds with robustness (Tsipras et al., 2019; Pang et al., 2022), in practice it is important that the condensed data enable the model to perform well both in i.i.d test setting and under adversarial attacks.

The most commonly used method to enhance robustness is adversarial training, which usually trains the model on the original label with the perturbed image. However, the semantic of the input image is sometimes changed by the perturbation even with the norm constraint, which may cause the cross-over mixture problem and severely degrade the clean accuracy (Zhang et al., 2020). Moreover, finding strong attacks often requires iterative optimization methods, which is computationally expensive.

On the other hand, if we choose to directly optimize for the robust dataset condensation objective, the tri-level optimization problem will result in a hugely inefficient process.

We instead find solution based on our theoretical discussion in Section 5. Although the smoothness of the loss landscape is related to both the gradient and curvature, previous work has shown that regularizing gradients gives a false sense of security about the robustness of neural networks (Athalye et al., 2018). With a vanished or obfuscated gradient around the data points, the model is still inherently robust to small perturbations. Therefore we regularize the curvature term $\lambda_1$ in our method, GUARD.

**Curvature Regularization** To reduce $\lambda_1$ in Eq. 3 requires computing the Hessian matrix and get the largest eigenvalue $\lambda_1$, which is quite expensive. Here we find an efficient approximation of it. Let $\mathbf{v_1}$ be the unit eigenvector corresponding to $\lambda_1$, then the Hessian-vector product

$$\mathbf{Hv_1} = \lambda_1 \mathbf{v_1} = \lim_{h \to 0} \frac{\nabla \ell(\mathbf{x} + h\mathbf{v_1}) - \nabla \ell(\mathbf{x})}{h} \tag{6}$$

We take the differential approximation of the Hessian-vector product, because we are interested in the curvature in a local area of $x$ rather than its asymptotic property. Therefore, for a small $h$,

$$\lambda_1 = \|\lambda_1 \mathbf{v_1}\| \approx \|\frac{\nabla \ell(\mathbf{x} + h\mathbf{v_1}) - \nabla \ell(\mathbf{x})}{h}\| \tag{7}$$

Previous work (Fawzi et al., 2018; Jetley et al., 2018; Moosavi-Dezfooli et al., 2019) has empirically shown that the direction of the gradient has a large cosine similarity with the direction of $\mathbf{v_1}$ in the input space of neural networks. Instead of calculating $\mathbf{v_1}$ directly, it is more efficient to take the gradient direction as a surrogate of $\mathbf{v_1}$ to perturb the input $\mathbf{x}$. So we replace the $\mathbf{v_1}$ above with the

normalized gradient $\mathbf{z} = \frac{\nabla \ell(\mathbf{x}))}{\|\nabla \ell(\mathbf{x}))\|}$, and define the regularized loss $\ell_R$ to encourage linearity in the input space:

$$\ell_R(\mathbf{x}) = \ell(\mathbf{x}) + \lambda \|\nabla \ell(\mathbf{x} + h\mathbf{z}) - \nabla \ell(\mathbf{x})\|^2 \tag{8}$$

where $\ell$ is the original loss function, $h$ is the discretization step, and the denominator is merged to the regularization coefficient.

**Gradient Matching**  Given our discussion about the connection between robustness and curvature, one may intuitively think that we can condense a robust dataset by matching its distribution with that of the real data using a feature extractor with reduced curvature. However, this may not be the case. While a robust dataset should be distributionally close to real datasets under robust feature extractors, the converse may be not true, i.e., such a dataset does not necessarily equip a model with robustness. Our experiment results have verified this in Appendix B.

Instead, we look for another solution. We argue that a robust dataset should be one that leads the model to a parameter value with good accuracy and robustness via standard training by encouraging the model to have a low curvature. Inspired by recent gradient-matching based methods, we hypothethize that a robust dataset should produce similar gradients as a model under robust training, and that the converse is also true. This motivates our method of regularizing curvature during gradient matching-based condensation.

Given a real dataset $\mathcal{T}$, the original DC formulation iteratively minimizes the distance between the gradient of the loss function on $\mathcal{T}$ and the synthetic dataset $\mathcal{S}$ through some distance function $D$, for a total of $T$ iterations:

$$\min_{\mathcal{S}} E_{\boldsymbol{\theta}_0 \sim P_{\boldsymbol{\theta}_0}} \left[ \sum_{t=0}^{T-1} D(\nabla_\theta \mathcal{L}^{\mathcal{T}}(\boldsymbol{\theta}_t), \nabla_\theta \mathcal{L}^{\mathcal{S}}(\boldsymbol{\theta}_t)) \right] \tag{9}$$

In the previous section, we have demonstrated that reducing the curvature of the loss function can lead to reduced adversarial loss. Let $\mathbf{H}$ be the Hessian matrix at some data point, our goal is to limit the largest eigenvalue $\lambda_1$ of the synthetic dataset. To do this, we use the curvature regularization method (Moosavi-Dezfooli et al., 2019). Consider the regularizer $\mathcal{R} = \|\nabla \ell(x + hz) - \nabla \ell(x)\|^2$. We define the regularized loss on the real dataset as $\mathcal{L}_R = \ell + \lambda \mathcal{R}$, where $\lambda$ is some constant that determines the strength of regularization. Then, we have the formulation for GUARD as

$$\min_{\mathcal{S}} E_{\boldsymbol{\theta}_0 \sim P_{\boldsymbol{\theta}_0}} \left[ \sum_{t=0}^{T-1} D(\nabla_\theta \mathcal{L}_R^{\mathcal{T}}(\boldsymbol{\theta}_t), \nabla_\theta \mathcal{L}^{\mathcal{S}}(\boldsymbol{\theta}_t)) \right] \tag{10}$$

We showcase the exact algorithm of GUARD in Appendix C.

## 7 EXPERIMENTS

### 7.1 EXPERIMENT SETTINGS

We present classification results on two popular datasets: MNIST and CIFAR10. We train networks using synthetic datasets and verify the network's performance on the original datasets.

We use the same ConvNet architecture (Gidaris & Komodakis, 2018) across the two datasets. The ConvNet contains 3 layers, with each layer using 128 filters, instance normalization (Ulyanov et al., 2016), ReLU activation function, and average pooling. All network weights are initialized using He Initialization (He et al., 2015).

We learn each synthetic dataset using 1000 iterations, with 256 images per training batch. We set the learning rate $\eta_\theta$ for updating network weights as 0.01 and the learning rate $\eta_S$ for updating synthetic dataset as 0.1. We use a different number of steps to update the network weights ($\varsigma_\theta$) and synthetic dataset ($\varsigma_S$) depending on the condensation scale. At 1 image per class (ipc), we set $\varsigma_\theta = 1$ and $\varsigma_S = 1$. At 10 ipc, we set $\varsigma_\theta = 50$ and $\varsigma_S = 10$. And at 50 ipc, we set $\varsigma_\theta = 10$ and $\varsigma_S = 50$.

For evaluation, we train a randomly initialized network with synthetic dataset for 1000 epochs with a learning rate of 0.01. We repeat each evaluation 50 times and report the mean and standard deviation.

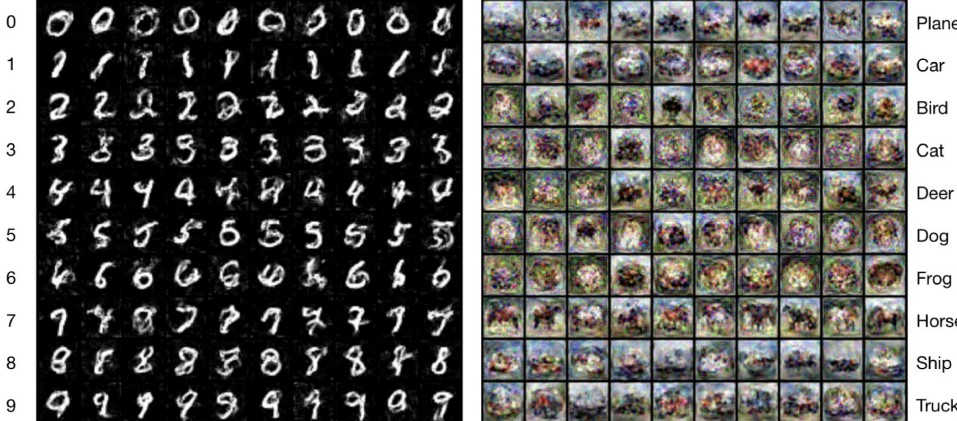

Figure 2: Visualization of synthetic datasets generated using GUARD with 10 images per class from MNIST and CIFAR10 datasets.

## 7.2 COMPARISON WITH OTHER METHODS

We compare our method to a comprehensive robustness benchmark for a large number of SOTA methods in dataset distillation, including DD (Wang et al., 2018), DC (Zhao et al., 2021), DSA (Zhao & Bilen, 2021), DM (Zhao & Bilen, 2023), MTT (Cazenavette et al., 2022), and FrePo (Zhou et al., 2022). We evaluate every method on each dataset with two condensation scales: 1 ipc and 10 ipc.

We conducted three set of attacks on each setting: $\ell_\infty *$, $\ell_\infty$, and $\ell_2$. The exact setting of each set of attack differ depending on the dataset. $\ell_\infty *$ is the strongest set of attack. For MNIST, the parameters are $\epsilon = 0.3$, $\alpha = 0.1$, steps $= 10$. For CIFAR10, the parameters are $\epsilon = 8/255$, $\alpha = 2/255$, steps $= 10$. $\ell_\infty$ is a weaker version of the previous attack, with $\epsilon = 0.1$, $\alpha = 0.03$, steps $= 10$ for MNIST and $\epsilon = 2/255$, $\alpha = 0.5/255$, steps $= 10$ for CIFAR10. Finally, $\ell_2$ is similar to $\ell_\infty$ in intensity, except that it uses $\ell_2$ distance, and its parameters are $\epsilon = 1$, $\alpha = 0.3$, steps $= 10$ for MNIST and $\epsilon = 0.2$, $\alpha = 0.05$, steps $= 10$ for CIFAR10.

## 7.3 RESULTS

Table 2: Robust accuracies of models trained on synthetic datasets from different methods, under various attack settings.

| | ipc | Attack | DD | DC | DSA | Method
DM | MTT | FrePo | Ours |
|---|---|---|---|---|---|---|---|---|---|
| MNIST | 1 | $\ell_\infty *$ | - | 0.3±0.2 | 0.2±0.1 | 0.2±0.2 | 0.0±0.0 | 0.0±0.0 | **0.7±0.2** |
| | | $\ell_\infty$ | - | 50.0±1.8 | 50.4±1.8 | 52.0±1.9 | 44.6±1.9 | 36.2±3.8 | **55.6±0.2** |
| | | $\ell_2$ | - | 63.6±1.1 | 63.5±1.0 | 65.8±1.1 | 62.2±1.9 | 59.1±2.3 | **66.2±0.5** |
| | 10 | $\ell_\infty *$ | 0.0±0.0 | 0.0±0.0 | 0.0±0.0 | 0.0±0.0 | 0.0±0.0 | 0.0±0.0 | **0.5±0.2** |
| | | $\ell_\infty$ | 34.3±4.1 | 68.0±1.9 | 71.2±1.6 | 65.3±2.7 | 63.7±2.0 | 54.3±3.7 | **74.6±0.5** |
| | | $\ell_2$ | 52.1±2.6 | 80.5±0.8 | 81.5±0.6 | 78.6±1.2 | 74.3±1.3 | 74.4±2.0 | **81.8±0.4** |
| CIFAR10 | 1 | $\ell_\infty *$ | - | 2.1±0.4 | 3.1±0.5 | 2.1±0.7 | 0.1±0.1 | 0.4±0.3 | **4.7±0.3** |
| | | $\ell_\infty$ | - | 17.5±0.6 | 17.7±0.6 | 14.6±0.9 | 10.3±1.0 | 7.6±0.8 | **19.3±0.4** |
| | | $\ell_2$ | - | 21.8±0.5 | 22.0±0.4 | 18.8±0.6 | 17.0±1.1 | 11.2±0.9 | **22.5±0.2** |
| | 10 | $\ell_\infty *$ | 1.0±0.3 | 0.3±0.1 | 0.3±0.1 | 0.3±0.1 | 0.1±0.0 | 0.9±0.6 | **1.9±0.3** |
| | | $\ell_\infty$ | 14.3±0.7 | 18.8±0.9 | 17.7±0.8 | 14.8±0.8 | 15.5±0.8 | 12.6±0.6 | **23.5±0.4** |
| | | $\ell_2$ | 18.6±0.6 | 27.2±0.8 | 26.1±0.7 | 22.4±1.0 | 27.0±0.8 | 17.2±0.7 | **28.2±0.2** |

We provide a visualization of the synthetic dataset generated by GUARD in Figure 2, utilizing a condensation scale of 10 images per class. While some images bear recognizable resemblances to objects within their respective class, numerous images appear to be less distinguishable. Notably,

classes possessing more distinctive outlines, such as horses, are far more recognizable than those with simpler outlines, such as frogs, in the synthetic dataset.

The results of our experiment are shown in Table 2. It can be observed that our method, in all dataset and attack settings, achieves the best performance. For instance, for the 10 ipc setting with CIFAR10, GUARD shows an improvement of 1%, 4.7%, and 1% over the best performance among all other methods, under the three sets of attack parameters, respectively. We also show a comparative analysis between the clean accuracies of GUARD and other dataset condensation methods in Appendix D. Generally, a model trained on GUARD is able to achieve clean accuracies that are almost at the same level as DC.

Comparing the performance of GUARD to a benchmark of non-robust methods doesn't fully demonstrate its effectiveness. To provide a more complete evaluation, we also show that GUARD is not only more efficient to train but also achieves better results when compared to adversarial training on condensed datasets, as detailed in Appendix E. We believe our edge over adversarial training, which is recognized as one of the most effective methods in real dataset settings, underscores the non-trivial effectiveness of GUARD.

We also evaluated GUARD against a variety of other adversarial attacks to further study its capability to enhance adversarial robustness in Appendix F. The adversarial attacks employed include PGD100, Square, and AutoAttack. This assortment includes both white-box and black-box attacks, providing a multifaceted evaluation of GUARD. The experimental outcomes demonstrate that the adversarial robustness provided by GUARD extends well to a diverse array of attacks.

### 7.4 TRANSFERABILITY TO OTHER METHODS

Since GUARD is applied on top of another dataset condensation method, it is no surprise that GUARD will inherit many limitations of that method. For example, it has been noted that DC (Zhao et al., 2021) exhibits a limitation with high-resolution datasets like ImageNet, thereby impacting the effectiveness of GUARD when DC is utilized as the underlying method.

However, we show that GUARD is able to be transferred to alternative dataset condensation methods. Such transferability is feasible as long as the alternative method utilizes a network trained on the real dataset as a comparison target during the condensation process, which is very common among dataset condensation methods. To demonstrate this, we extended GUARD to a recent technique called SRe$^2$L (Yin et al., 2023), which performs much better on ImageNet than DC. The comparison between vanilla SRe$^2$L and SRe$^2$L augmented with GUARD revealed that GUARD notably improves the robustness of the method, indicating its effectiveness towards high-resolution datasets and proving its flexibility to be incorporated with other condensation methods. Detailed results of this analysis are provided in Appendix G.

## 8 CONCLUSIONS

Our work focuses on a novel perspective on dataset condensation, emphasizing its robustness characteristics. While there are reasons that might suggest dataset condensation inherently boosts the robustness of trained models, our empirical experiments indicate otherwise. This discovery prompted us to delve deeper into understanding and resolving this issue. Through our investigation, we proposed a theory for robust dataset condensation, deriving a significant insight: the optimization of robustness with respect to synthetic and real datasets is differentiated only by a constant term. This conclusion opens up various potentials for subsequent research in the field. Moreover, our GUARD method has demonstrated effectiveness in enhancing robustness against diverse types of attacks. It is also potentially applicable to numerous dataset condensation techniques, not limited to the DC method exclusively. We have also established a robustness benchmark through extensive experimentation with common dataset condensation methods, uncovering their respective robustness levels. Looking forward, it is our hope that this work will inspire and support further research and development in this area. We hope our work contributes to the development of dataset condensation techniques that are not only efficient but also robust. Such advancements would broaden the applicability of condensed datasets across many more settings.

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
