## A PROOF OF PROPOSITION 1

The adversarial loss of an arbitrary input sample $\mathbf{x}$ can be upper-bounded as below

$$\tilde{\ell}_\rho^{adv}(\mathbf{x}) = \max_{\|\mathbf{v}\|\leq\rho} \tilde{\ell}(\mathbf{x}+\mathbf{v}) = \max_{\|\mathbf{v}\|\leq\rho} \ell(\mathbf{x}) + \nabla\ell(\mathbf{x})^\top\mathbf{v} + \frac{1}{2}\mathbf{v}^\top\mathbf{H}\mathbf{v} \tag{1}$$

$$\leq \max_{\|\mathbf{v}\|\leq\rho} \ell(\mathbf{x}) + \|\nabla\ell(\mathbf{x})\|\|\mathbf{v}\| + \frac{1}{2}\lambda_1(\mathbf{x})\|\mathbf{v}\|^2 \tag{2}$$

$$= \ell(\mathbf{x}) + \|\nabla\ell(\mathbf{x})\|\rho + \frac{1}{2}\lambda_1(\mathbf{x})\rho^2 \tag{3}$$

where $\lambda$ is the largest eigenvalue of the Hessian $\mathbf{H}(\ell(\mathbf{x}))$.
Taking expectation over the distribution of real data with class label $c$, denoted as $D_c$

$$\mathbb{E}_{\mathbf{x}\sim D_c} \tilde{\ell}_\rho^{adv}(\mathbf{x}) \leq \mathbb{E}_{\mathbf{x}\sim D_c} \ell(\mathbf{x}) + \rho \mathbb{E}_{\mathbf{x}\sim D_c} \|\nabla\ell(\mathbf{x})\| + \frac{1}{2}\rho^2 \mathbb{E}_{\mathbf{x}\sim D_c} \lambda_1(\mathbf{x}) \tag{4}$$

With the assumption that $\tilde{\ell}(\mathbf{x})$ is convex, we know that $\tilde{\ell}_\rho^{adv}(\mathbf{x})$ is also convex, because $\forall \lambda \in [0,1]$,

$$\tilde{\ell}_\rho^{adv}(\lambda\mathbf{x}_1 + (1-\lambda)\mathbf{x}_2) \tag{5}$$

$$= \max_{\|\mathbf{v}\|\leq\rho} \tilde{\ell}(\lambda\mathbf{x}_1 + (1-\lambda)\mathbf{x}_2 + \mathbf{v}) \tag{6}$$

$$= \max_{\|\mathbf{v}\|\leq\rho} \tilde{\ell}(\lambda(\mathbf{x}_1+\mathbf{v}) + (1-\lambda)(\mathbf{x}_2+\mathbf{v})) \tag{7}$$

$$\leq \max_{\|\mathbf{v}\|\leq\rho} \lambda\tilde{\ell}(\mathbf{x}_1+\mathbf{v}) + (1-\lambda)\tilde{\ell}(\mathbf{x}_2+\mathbf{v}) \tag{8}$$

$$\leq \lambda \max_{\|\mathbf{v}\|\leq\rho} \tilde{\ell}(\mathbf{x}_1+\mathbf{v}) + (1-\lambda) \max_{\|\mathbf{v}\|\leq\rho} \tilde{\ell}(\mathbf{x}_2+\mathbf{v}) \tag{9}$$

$$= \lambda\tilde{\ell}_\rho^{adv}(\mathbf{x}_1) + (1-\lambda)\tilde{\ell}_\rho^{adv}(\mathbf{x}_2) \tag{10}$$

Therefore, by Jensen's Inequality

$$\tilde{\ell}_\rho^{adv}(\mathbb{E}_{\mathbf{x}\sim D_c} \mathbf{x}) \leq \mathbb{E}_{\mathbf{x}\sim D_c} \tilde{\ell}_\rho^{adv}(\mathbf{x}) \tag{11}$$

Let $\mathbf{x}'$ be a datum distilled from the training data with class label $c$. It should be close in distribution to that of the real data. From the MMD loss used by DM (Zhao & Bilen, 2023), we assume that $\|h(\mathbf{x}') - \mathbb{E}_{\mathbf{x}\sim D_c}h(\mathbf{x})\| \leq \sigma$, where $h(\cdot)$ is a feature extractor. If $h(\cdot)$ is invertible, then $\mathcal{L}_\rho^{adv}(\cdot) = \tilde{\ell}_\rho^{adv}(h^{-1}(\cdot))$ is a function defined on the feature space. We assume that $\mathcal{L}_\rho^{adv}(\cdot)$ is $L$-Lipschitz, it follows that

$$\mathcal{L}_\rho^{adv}(h(\mathbf{x}')) \leq \mathcal{L}_\rho^{adv}(\mathbb{E}_{\mathbf{x}\sim D_c} h(\mathbf{x})) + L\sigma \tag{12}$$

If we add the assumption that $h(\cdot)$ is linear, $\mathbb{E}_{\mathbf{x}\sim D_c} h(\mathbf{x}) = h(\mathbb{E}_{\mathbf{x}\sim D_c} \mathbf{x})$, then

$$\tilde{\ell}_\rho^{adv}(\mathbf{x}') \leq \tilde{\ell}_\rho^{adv}(\mathbb{E}_{\mathbf{x}\sim D_c} \mathbf{x}) + L\sigma \tag{13}$$

Combining Eq. 4, 11, 13, we get

$$\tilde{\ell}_\rho^{adv}(\mathbf{x}') \leq \mathbb{E}_{\mathbf{x}\sim D_c} \ell(\mathbf{x}) + \rho \mathbb{E}_{\mathbf{x}\sim D_c} \|\nabla\ell(\mathbf{x})\| + \frac{1}{2}\rho^2 \mathbb{E}_{\mathbf{x}\sim D_c} \lambda_1(\mathbf{x}) + L\sigma \tag{14}$$

**Discussion** The inequality in line (2) is an equality if and only if the direction of the gradient is the same as the direction of $\lambda_1$. Previous work has empirically shown that the two directions have a large cosine similarity in the input space of neural networks. Our assumption about the Lipschitz continuity of $\mathcal{L}_\rho^{adv}(\cdot)$ is reasonable, as recent work has shown improved estimation of the Lipschitz constant of

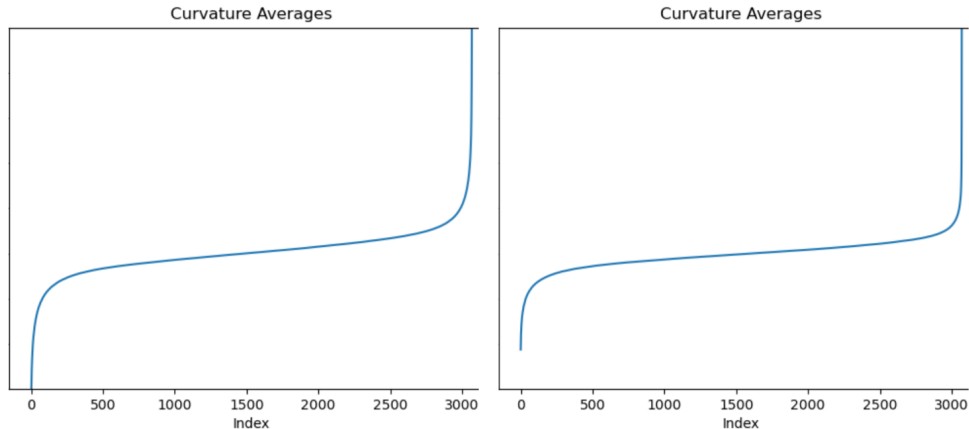

Figure 3: A comparison between the curvature profiles of DC (left) and GUARD (right), which is shown in the form of sorted eigenvalues of the Hessian. We took the average of sorted eigenvalues of 100 samples in the real training data.

neural networks in a wide range of settings (Khromov & Singh, 2023). Although our assumptions about the convexity of $\tilde{\ell}(\mathbf{x})$ and the linearity of $h(\cdot)$ is relatively strong, it still reflects important aspects of reality, as our experiment in Table 2 has shown that reducing the curvature term in r.h.s of Eq. 14 effectively improves the robustness of models trained on distilled data. Moreover, in Fig. 3 we plot the distribution of eigenvalues of the real data samples on the loss landscape of a model trained on standard distilled data and a model trained on robust distilled data from our GUARD method, respectively. GUARD corresponds to a flatter curve of eigenvalue distribution, indicating that the loss landscape becomes more linear after our regularization.

We show the curvature profiles of DC and GUARD to demonstrate that GUARD does limit the curvature profile of the resulting dataset in Figure 3. It can be seen that GUARD (right) has much lower curvature profiles towards both ends of the plot.

## B  EXPERIMENT RESULTS ON ROBUST DISTRIBUTION MATCHING

We experimented with Dataset Condensation with Distribution Matching (DM) (Zhao & Bilen, 2023) using robust feature extractors. We used a pool of 50 ConvNet models with different initialization, and one model is randomly selected as the feature extractor at each iteration. Two settings were used: in the online learning setting (denoted as "DM+O"), we alternated between training the synthetic set with the MMD loss and training the model with the curvature regularization; in the pretraining setting (denoted as "DM+P"), we pretrained all the models with the curvature regularization on the real dataset for 10 epochs before starting data distillation. The result is shown in Table 3. With the large computational cost of pretraining, the "DM+P" setting only brings marginal and inconsistent improvement over the DM method, and the "DM+O" setting degrades the performance severely.

Table 3: Robust accuracies of models trained on synthetic datasets from DM, under various attack settings.

| | | MNIST | | | | | | CIFAR10 | | | | | |
|---|---|---|---|---|---|---|---|---|---|---|---|---|---|
| | ipc | | 1 | | | 10 | | | 1 | | | 10 | |
| | Attack | $\ell_\infty$* | $\ell_\infty$ | $\ell_2$ | $\ell_\infty$* | $\ell_\infty$ | $\ell_2$ | $\ell_\infty$* | $\ell_\infty$ | $\ell_2$ | $\ell_\infty$* | $\ell_\infty$ | $\ell_2$ |
| Method | DM | 0.2±0.2 | 52.0±1.9 | 65.8±1.1 | 0.0±0.0 | 65.3±2.7 | 78.6±1.2 | 2.1±0.7 | 14.6±0.9 | 18.8±0.6 | 0.3±0.1 | 14.8±0.8 | 22.4±1.0 |
| | DM + P | 0.8±0.3 | 52.9±2.8 | 63.9±1.3 | 0.0±0.0 | 66.9±2.6 | 78.6±1.5 | 3.7±0.8 | 16.0±0.5 | 19.5±0.8 | 0.3±0.1 | 17.2±0.8 | 25.4±0.7 |
| | DM + O | 0.9±0.3 | 45.1±1.8 | 57.1±1.5 | 0.0±0.0 | 68.4±2.3 | 79.8±1.2 | 0.2±0.2 | 5.8±1.2 | 9.7±1.1 | 0.1±0.0 | 12.7±0.8 | 20.5±0.7 |

## C    ALGORITHM OF GUARD

We present the exact algorithm of GUARD in Algorithm 1. For each outer iteration $k$, we sample a new initial weight from some random distribution of weights to ensure the synthetic dataset can generalize well to a range of weight initializations. After, we iteratively sample a minibatch pair from the real dataset and the synthetic dataset and compute the loss over them on a neural network with the weights $\theta_t$. We compute the regularized loss on real data through Eq. 8. Finally, we compute the gradient of the losses w.r.t. $\theta$, and update the synthetic dataset through stochastic gradient descent on the distance between the gradients. At the end of each inner iteration $t$, we update the weights $\theta_{t+1}$ using the updated synthetic dataset.

---

**Algorithm 1:** GUARD (Geometric regUlarization for Adversarial Robust Dataset)

---

**Input:** $\mathcal{T}$: Training set; $S$: initial synthetic dataset with $C$ classes; $p(\theta_0)$: initial weights distribution; $\phi_\theta$: neural network; $K$: number of outer-loop steps; $T$: number of inner-loop steps; $\varsigma_\theta$: number of steps for updating weights; $\varsigma_S$: number of steps for updating synthetic samples; $\eta_\theta$: learning rate for updating weights; $\eta_S$: learning rate for updating synthetic samples; $D$: gradient distance function; $h$: discretization step; $\lambda$: strength of regularization

**for each** outer training step $k = 1$ to $K$ **do**

    Sample initial weight $\theta_0 \sim p(\theta_0)$

    **for each** inner training step $t = 1$ to $T$ **do**

        **for each** class $c = 1$ to $C$ **do**

            Sample $\omega \sim \Omega$ and a minibatch pair $B_c^{\mathcal{T}} \sim \mathcal{T}$ and $B_c^{S} \sim S$

            Compute loss on synthetic data $\mathcal{L}_c^{S} = \frac{1}{|B_c^{S}|} \sum_{(\mathbf{s},\mathbf{y}) \in B_c^{S}} \ell(\phi_{\theta_t}(\mathbf{s}), \mathbf{y})$

            Compute loss on real data $\mathcal{L}_c^{\mathcal{T}} = \frac{1}{|B_c^{\mathcal{T}}|} \sum_{(\mathbf{x},\mathbf{y}) \in B_c^{\mathcal{T}}} \ell(\phi_{\theta_t}(\mathbf{x}), \mathbf{y})$

            Compute $z = \frac{\nabla\ell(\phi_{\theta_t}(\mathbf{s}),\mathbf{y})}{\|\nabla\ell(\phi_{\theta_t}(\mathbf{s},\mathbf{y}))\|}$

            Compute loss on perturbed real data $\mathcal{L}_c^{\mathcal{T}_z} = \frac{1}{|B_c^{\mathcal{T}}|} \sum_{(\mathbf{x},\mathbf{y}) \in B_c^{\mathcal{T}}} \ell(\phi_{\theta_t}(\mathbf{x} + hz), \mathbf{y})$

            Compute regularizer $\mathcal{R} = \nabla_\theta \mathcal{L}_c^{\mathcal{T}_z}(\theta_t) - \nabla_\theta \mathcal{L}_c^{\mathcal{T}}(\theta_t)$

            Compute regularized loss on real data $\mathcal{L}_c^{\mathcal{T}_{\mathcal{R}}} = \mathcal{L}_c^{\mathcal{T}} + \lambda\mathcal{R}$

            Update $S_c \leftarrow \mathtt{sgd}_S(D(\nabla_\theta \mathcal{L}_c^{S}(\theta_t), \nabla_\theta \mathcal{L}_c^{\mathcal{T}_{\mathcal{R}}}(\theta_t)), \varsigma_S, \eta_S)$

        **end**

        Update $\theta_{t+1} \leftarrow \mathtt{sgd}_\theta(\mathcal{L}(\theta_t, S), \varsigma_\theta, \eta_\theta)$

    **end**

**end**

**Output:** robust condensed dataset $S$

---

## D    CLEAN ACCURACY OF THE DISTILLATION METHODS

In addition to conducting various experiments on robustness, we also examined how GUARD affects the clean accuracy of the trained model. In Table 4, we present the performance of our method in standard i.i.d. setting in comparison with other methods.

Table 4: Clean accuracy of models trained on synthetic datasets from different methods

| | ipc | Method | | | | | | |
| | | DD | DC | DSA | DM | MTT | FrePo | Ours |
|---|---|---|---|---|---|---|---|---|
| MNIST | 1 | / | 86.5±0.7 | 86.3±0.7 | 89.7±0.6 | 91.4±0.9 | **93.0±0.4** | 89.2±0.3 |
| | 10 | 79.5±8.1 | 96.4±0.1 | 96.1±0.1 | 97.5±0.1 | 97.3±0.1 | **98.6±0.1** | 95.8±0.1 |
| CIFAR10 | 1 | / | 29.9±0.7 | 29.5±0.6 | 26.0±0.8 | 46.3±0.8 | **46.8±0.7** | 30.9±0.4 |
| | 10 | 36.8±1.2 | 45.3±0.5 | 43.5±0.5 | 48.9±0.6 | 65.3±0.7 | **65.5±0.4** | 46.8±0.4 |

# E    COMPARISON BETWEEN ADVERSARIAL TRAINING AND GUARD

In Table 5, we show a comparison between using adversarial training and using GUARD for dataset condensation. For the adversarial training process, a dataset is initially condensed using the DC method. Then, while a model is training on this synthetic dataset, an adversarial attack is applied to the synthetic images following standard adversarial training procedures. In this case, the adversarial attack employed is a PGD attack under the $\ell_\infty$ setting as detailed in the Experiments section of this paper.

Table 5: A comparison between training time, clean accuracy and robust accuracy of models trained on DC with adversarial training and GUARD

|  | ipc | DC + Adversarial Training | | | GUARD | | |
|---|---|---|---|---|---|---|---|
|  |  | Training Time | Clean Acc | Robust Acc | Training Time | Clean Acc | Robust Acc |
| MNIST | 1 | 15.2±0.3s | 81.5±0.8 | 0.5±0.2 | **2.1±0.2s** | **89.2±0.3** | **0.7±0.3** |
|  | 10 | 24.3±0.1s | 95.5±0.1 | 0.0±0.0 | **2.9±0.3s** | **95.8±0.1** | **0.5±0.2** |
| CIFAR10 | 1 | 15.4±0.2s | 25.8±0.5 | 2.6±0.3 | **2.1±0.1s** | **30.9±0.4** | **4.7±0.3** |
|  | 10 | 24.7±0.1s | 34.4±0.6 | 0.7±0.2 | **3.0±0.1s** | **46.8±0.4** | **1.9±0.3** |

# F    OTHER ATTACKS ON GUARD

In Table 6, we compare some results from using other types of attacks against GUARD and DC. We used three different attacks, including PGD100 with parameters $\epsilon = 2/255$, $\alpha = 0.5/255$, steps $= 100$, Square Attack (Andriushchenko et al., 2020) with $\epsilon = 2/255$, and AutoAttack with $\epsilon = 2/255$.

Table 6: Robust accuracy of GUARD and DC on CIFAR10 under various attacks in 10 ipc setting

|  | PGD100 | Square | AutoAttack |
|---|---|---|---|
| GUARD | **15.8±2.0** | **15.7±2.1** | **13.7±1.8** |
| DC | 10.6±1.4 | 11.5±1.6 | 10.1±1.7 |

# G    TRANSFERABILITY OF GUARD

Table 7: Clean and robust accuracy of SRe$^2$L on ImageNette with and without GUARD, under PGD attack with $\epsilon = 2/255$, $\alpha = 0.5/255$, steps $= 100$

| ipc | Method | Clean Accuracy | $\ell_\infty$* | $\ell_\infty$ |
|---|---|---|---|---|
| 1 | SRe$^2$L | 24.2 | **2.0** | 9.6 |
|  | GUARD | **29.2** | 1.6 | **13.5** |
| 10 | SRe$^2$L | 51.4 | 0.0 | 6.7 |
|  | GUARD | **55.9** | **1.2** | **23.3** |

In this section, we discuss how GUARD can be transferred onto other dataset condensation methods. This is especially useful since different dataset condensation methods have their own advantages and drawbacks, therefore methods can be selected to best suit one's needs.

In essence, GUARD works by reducing the curvature on a network trained with real data, and optimizing the synthetic data to allow networks trained on it to have such characteristics. Therefore, GUARD can be theoretically applied to any dataset condensation method, given that it optimizes the synthetic dataset by comparing to a network trained with real data. This is very common among dataset condensation methods, including DD (Wang et al., 2018), DC (Zhao et al., 2021), DSA (Zhao & Bilen, 2021), MTT (Cazenavette et al., 2022), DCC (Lee et al., 2022), SRe$^2$L (Yin et al., 2023), and many more.

In this case, we used SRe$^2$L as an example, since it is a recent technique that performs much better than DC on ImageNet. SRe$^2$L works by first training a network with real data, then reconstructing images from the network, and finally cropping the images and assigning a soft-label to each crop with the network. GUARD can be applied to the first phase, allowing images to be reconstructed and labeled using a network with low curvature.

We used the $\ell_\infty$* and $\ell_\infty$ PGD attack settings, and used ImageNette, a selection of ten classes from ImageNet, as our dataset in this experiment. From the results in Table 7, it can be seen that GUARD helps SRe$^2$L to achieve significantly better robustness under adversarial attacks. In addition, possibly due to GUARD's nature as a regularizer, it also improved SRe$^2$L's clean accuracy as a side effect.