# OpenReview forum: "Towards Adversarially Robust Condensed Dataset by Curvature Regularization"
_ICLR.cc/2024/Conference — Submitted to ICLR 2024_

### Official Review · Reviewer_upET · 2023-10-25

**Soundness:** 2 fair
**Presentation:** 2 fair
**Contribution:** 2 fair
**Rating:** 5
**Confidence:** 2

**Summary:**

This paper highlights the vulnerability of existing dataset condensation methodologies to adversarial attacks and introduces a dataset condensation technique that takes adversarial robustness into account. The authors are motivated by the assertion that to ensure adversarial robustness, the data loss should exhibit a low curvature. Drawing inspiration from prior studies that enhanced adversarial robustness through curvature regularization, they bridge this approach with a gradient matching-based dataset condensation algorithm. As a result, they propose a methodology named "Geometric regUlarization for Adversarial Robust Dataset" (GUARD). The dataset produced using this methodology demonstrates superior adversarial robustness performance in various settings when compared to traditional techniques.

**Strengths:**

- The paper identifies the vulnerability of models trained using traditional data condensation methodologies to adversarial attacks, implying that condensed datasets might leverage spurious features and information.
- The authors point out that an adversarially trained model should possess a smooth loss landscape. They also highlight the challenges of adversarial training in dataset condensation, tackling the issue from a curvature perspective.
- By bridging two methodologies, gradient matching in data condensation and CURE in adversarial robustness, the paper introduces a robust data condensation method named GUARD. This proposed method integrates curvature considerations into the gradient matching process of condensed data via regularization.
- The results demonstrate that the proposed method achieves relatively higher robust accuracy compared to traditional methods. Furthermore, it displays superior performance and efficiency when compared with standard adversarial training techniques for dataset condensation.

**Weaknesses:**

- Previous methodologies were conducted without considerations related to adversarial robustness. Achieving performance that doesn't significantly surpass these methods might not hold substantial meaning. Notably, the performance against strong attacks does not seem to indicate significant robustness. When proposing a methodology considering adversarial robustness, I believe that the proposed method should exhibit a performance better than random chance (10% for 10-class tasks) when subjected to strong attacks. The setting for strong attacks reflects the benchmark for invisible perturbations in the adversarial robustness research area. Approaches that fail to demonstrate significant performance improvements in this setting are arguably limited.
- Naturally, training methodologies considering adversarial robustness demand higher computational costs compared to standard training. Given that the proposed method appears to simultaneously utilize both the conventional method (gradient matching) and curvature regularization, it seems to require a relatively high computational cost (please correct me if I'm mistaken). However, the enhanced robustness performance doesn't seem to justify this computational overhead.
- The proposed method shows a decline in performance on regular data compared to traditional methods. While a trade-off between robustness and accuracy can be expected, the drop in clean accuracy doesn't seem to be adequately compensated by an increase in robust accuracy.

**Questions:**

- Has there been an analysis or experiment regarding the computational cost, specifically on the overhead introduced compared to traditional methodologies?
- Are there performance results under the general AutoAttack setting, which has the same epsilon upper bound as the strong attack?
- In the weaknesses section, drawbacks are highlighted using the standard strong attack as a reference. What are the author's thoughts on this? My knowledge in the dataset condensation field is limited; hence, I'm unaware if there have been studies focusing on adversarial robustness in this domain. Considering the methodologies compared in the paper, it appears there might not be. Therefore, it's challenging to judge the appropriateness of the paper's settings. Given that the model has been trained on limited data, I'd like to inquire whether it's suitable to compare the performances of methodologies under a weaker setting (epsilon=2), rather than the conventional adversarial settings employed in the vision domain.
- Are there any other cases where adding curvature regularization to methods other than gradient matching leads to an improvement (not marginal) in robustness performance?

---

> ### Author Response · Authors · 2023-11-22
>
> We really appreciate your constructive feedback. In particular, we are happy that you recognized several strengths of our paper. Below are our responses to your questions and suggestions:
>
> **The proposed method should exhibit a performance better than random chance.**
>
> We don’t believe it is fair to hold adversarial defense methods for condensed datasets to such standard. Many of the adversarial defense methods that work well under the real dataset setting do not apply in the condensed dataset setting. This is because one of the fundamental premises of dataset condensation is that no real data will be given to the downstream model at training time, which is important to preserve many of the properties of condensed datasets, including size, training efficiency, and privacy. Without real data, the data distribution that the model learns is already largely different compared to the test images, making it much easier for the adversary to find adversarial examples. Therefore, we don’t think it is realistic to expect the same degree of robustness from condensed datasets and real datasets.
>
> In addition, one reason for the low level of robustness is the inherent low level of clean accuracy. For example, many adversarial defense methods can retain a robustness of around 50%, on a model that can normally achieve 90% clean accuracy on CIFAR10. However, DC can only achieve 50% clean accuracy on CIFAR10, so it is not surprising that when applied the same level of attack, the robustness accuracy deteriorates severely.
>
> We are aware of the lack of previous research in this area. This is why in addition to comparison to vanilla dataset condensation methods, in the appendix we also showed a comparison between our results and those obtained from adversarial training, which is recognized as one of the more effective adversarial defense methods for standard datasets. This comparison demonstrates that our method yields superior results.
>
> **The enhanced robustness performance doesn’t seem to justify this computational overhead. Has there been an analysis regarding computational cost?**
>
> Although we don’t have a formal analysis of the computational cost, it is very negligible. In fact, although the problem of robust dataset condensation is a tri-level optimization problem, we formulated our method in this exact way to reduce the potential computational cost by embedding the procedure in the bi-level optimization of dataset condensation.
>
> You can look at our algorithm in the appendix. Our perturbation $z$ is calculated from the gradients from the real images, where the gradient is already calculated in the original DC algorithm. Then, we simply do another forward pass (without a backward pass) with real images plus $z$, and calculate the final regularization through the difference of the losses. This means the overall computational overhead is only a single forward pass per iteration.
>
> **The proposed method shows decline in performance on regular data.**
>
> In fact, we show improvements to clean accuracy after applying our method in the appendix. (Our base condensation method is DC). While this increase in accuracy may be due to differences in experimental procedures, and is not an intended consequence of GUARD, it does show that there is no clear decrease in clean accuracy.
>
> **Are there performance results under AutoAttack setting with the same epsilon as strong attack?**
>
> Unfortunately we haven’t conducted experiments in that particular setting.
>
> **Are there any other cases where adding curvature regularization to methods other than gradient matching leads to an improvement?**
>
> Yes! In our appendix, we presented results where we utilized SRe2L [1], a condensation method that performs better on high resolution datasets, and tested our method on ImageNette where it showed an improvement.
>
> [1] Zeyuan Yin, Eric Xing, and Zhiqiang Shen. Squeeze, Recover and Relabel: Dataset Condensation at ImageNet Scale From A New Perspective. Neurips 2023.

---

### Official Review · Reviewer_8UKM · 2023-10-30

**Soundness:** 2 fair
**Presentation:** 3 good
**Contribution:** 3 good
**Rating:** 3
**Confidence:** 3

**Summary:**

This article proposes improving the adversarial robustness of models trained on compressed training data using the dataset condensation technique to accelerate training. The author first empirically verifies that dataset condensation alone does not enhance the model's adversarial robustness. Then, through theoretical derivation, they suggest regularizing the curvature term when performing dataset condensation. Finally, the author conducts experiments to demonstrate the effectiveness of their approach.

**Strengths:**

The topic studied in this paper is valuable and indeed offers a new direction for improving the adversarial robustness of models. The writing of the paper is also well-done, providing a detailed introduction to previous work. The paper includes both theoretical analysis and complementary experiments to support its claims. Overall, the paper is quite comprehensive.

**Weaknesses:**

However, I have some concerns regarding both the method and the experimental results. Firstly, looking at the experimental section, I'm not quite sure what the improvement in adversarial robustness at the scale shown in Table 2 indicates, especially when considering the standard deviation. It seems to only weakly suggest that the proposed method may exhibit better consistency in terms of adversarial robustness. I find it difficult to consider this experimental result as strong support for the author's conclusions.

Regarding the method, the author uses Taylor expansion to find the relationship between adversarial samples and the first and second-order information of samples under the model. However, they only briefly mention that "previous work has shown that regularizing gradients gives a false sense of security about the robustness of neural networks" and then solely rely on the curvature term as a regularization term -- without further theoretical or experimental justification. I find this unacceptable. Furthermore, the approximation of the Hessian matrix is very rough, and there is no further exploration or explanation regarding the parameter $h$.

Overall, the research direction is indeed valuable, but the current version is too rough and would be difficult to be considered as a top conference paper.

**Questions:**

Apart from the concerns about the method and its experimental results, I have some other questions as well.

Firstly, I am confused about the inequality sign in equation (2). If the adversarial samples are defined as maximizing the loss, then it seems that the inequality should be reversed. However, considering the higher-order terms, I am not entirely sure about the direction of this inequality. Please correct me if I misunderstood.

Secondly, I find the assumption in Proposition 1 regarding the convexity of $l(\cdot)$ confusing. Why can we make such an assumption?

Moving on to the experimental section, there is a significant decrease in performance on clean samples after applying this method (as shown in the appendix). I am unsure if this contradicts the statement mentioned in the paper that "in practice, it is important that the condensed data enable the model to perform well both in i.i.d test setting and under adversarial attacks" (currently, it seems contradictory). Additionally, these experimental results should be presented in the main body of the paper.

---

> ### Author Response · Authors · 2023-11-22
>
> We are very thankful for your constructive feedback. We especially appreciate your acknowledgment of the significance of our research topic and our effort to maintain clear writing. Below, we provide detailed responses to your questions and suggestions:
>
> **The result seems to only weakly suggest that the proposed method may exhibit better consistency in terms of adversarial robustness.**
>
> We do agree the results may seem weak, but improving adversarial robustness on condensed datasets is not a trivial task. Many of the adversarial defense methods that work well under the real dataset setting do not apply in the condensed dataset setting. This is because one of the fundamental premises of dataset condensation is that no real data will be given to the downstream model at training time, which is important to preserve many of the properties of condensed datasets, including size, training efficiency, and privacy. Without real data, the data distribution that the model learns is already largely different compared to the test images, making it much easier for the adversary to find adversarial examples. Therefore, we don’t think it is realistic to expect the same degree of robustness from condensed datasets and real datasets.
>
> In addition, one reason for the low level of robustness is the inherent low level of clean accuracy. For example, many adversarial defense methods can retain a robustness of around 50%, on a model that can normally achieve 90% clean accuracy on CIFAR10. However, DC can only achieve 50% clean accuracy on CIFAR10, so it is not surprising that when applied the same level of attack, the robustness accuracy deteriorates severely.
>
> In the appendix, we conducted a comparison between our results and those obtained from adversarial training, which is recognized as one of the more effective adversarial defense methods for standard datasets. This comparison demonstrates that our method yields superior results.
>
>
> **The authors only briefly mention that “previous work has shown that regularizing gradients give a false sense of security”**
>
> Our proposition highlights that both gradients and curvature impact the upper bound of the robust loss function. We are aware of existing research indicating that gradient-based defenses can create a false sense of security and are vulnerable to certain attacks, so we conducted preliminary experiments that suggested gradient regularization generally underperforms compared to curvature regularization. Based on these findings, we chose to focus on curvature regularization in our paper.
>
> **I am unsure if this contracts the statement that “it is important that the condensed data enable the model to perform well both in i.i.d test setting and under adversarial attacks**
>
> The condensation method employed in our study is DC, and in the appendix we actually demonstrated improvements in clean accuracy after applying our method. Although this increase in accuracy might partially result from differences in experimental procedures, it importantly indicates that our approach does not lead to a clear decrease in clean accuracy.

---

> > ### Comment · Reviewer_8UKM · 2023-11-22
> >
> > Thank you for the response. Unfortunately, my concern has not been fully addressed, so I will not consider raising the score at this time. I can see the significance of this direction, but the current version of the paper is unlikely to be considered as top conference work. I hope the authors can redesign and conduct experiments in the future, providing strong evidence to demonstrate the effectiveness of the proposed method.

---

### Official Review · Reviewer_r5Wk · 2023-10-31

**Soundness:** 1 poor
**Presentation:** 1 poor
**Contribution:** 1 poor
**Rating:** 1
**Confidence:** 4

**Summary:**

This paper studies the adversarial robustness property of a recent technique called Dataset Condensation (DC). DC learns to synthesize a smaller dataset such that models can still have good generalization performance if they are trained on this smaller dataset. The authors provided a negative answer to the hypothesis -- DC can generate datasets that can foster adversarially robust models by discarding non-observable features that often lead to adversarial vulnerability. Finally, the authors proposed a method called GUARD (Geometric regUlarization for Adversarial Robust Dataset) to improve adversarial robustness for DC methods by incorporating curvature regularization. The effectiveness of the proposed method is verified by some experiments on MNIST and CIFAR10 classification tasks.

**Strengths:**

+ The adversarial robustness property of DC methods has not been studied before.

+ The authors first clarified that adversarial robustness is not held for DC and then proposed a method to improve such property.

**Weaknesses:**

- Dataset Condensation (DC) was a recent technique that has not been recognized by many people. Whether it is promising to be used in the big data era is not clear. The authors proposed to study whether DC methods are adversarially robust. The motivation is very weak. I cannot see any significance in studying this topic.

- Although overturned by the authors, the hypothesis "adversarial robustness might be inherently induced by the dataset condensation process" does not make much sense. It is not clear why DC can be linked with the previous work that suggested the adversarial vulnerability of models is often tied to these non-observable features since the authors did not study what type of features are discarded by DC methods. Also, the goal of DC seems to reduce training samples other than discarding non-observable features. The connection between these two types of work seems to be hallucinated without any evidence.

- The novelty of the proposed method is not clear. Curvature Regularization has been considered by (Moosavi-Dezfooli et al. 2019). The technical novelty is limited.

- The experiment is weak. Only conduct on MNIST and CIFAR10.  Only evaluate ConvNet with 3 layers.

- The authors only compare the adversarial robustness performance with several DC methods. The improvements are very marginal. For example, for l-inf attack, the performance of the proposed method is 0.7±0.2, which can not show any improvements. In other words, the proposed method also fails under this attack.

**Questions:**

What does Figure 1 mean?

What is the function of Figure 2?

---

> ### Author Response · Authors · 2023-11-22
>
> We sincerely appreciate your constructive feedback. We are particularly grateful for your recognition of our pioneering efforts in studying this specific topic. Below, we present our responses to your questions and suggestions:
>
> **DC was a recent technique that has not been recognized by many people.**
>
> Dataset condensation has been proven of interest to the community by its popularity and the large body of related works. Our work is extremely relevant in this area, as several papers [1, 2] have discussed the potential application of dataset condensation in trustworthy machine learning (and hence why we study the topic of adversarial robustness).
>
> [1] Zongxiong Chen, Jiahui Geng, Derui Zhu, Herbert Woisetschlaeger, Qing Li, Sonja Schimmler, Ruben Mayer, and Chunming Rong. A Comprehensive Study on Dataset Distillation: Performance, Privacy, Robustness and Fairness. arXiv preprint arXiv:2305.03355, 2023.
>
> [2] Jiahui Geng, Zongxiong Chen, Yuandou Wang, Herbert Woisetschlaeger, Qing Li, Sonja Schimmler, Ruben Mayer, Zhiming Zhao, and Chunming Rong. A Survey on Dataset Distillation: Approaches, Applications, and Future Directions. arXiv preprint arXiv:2305.01975, 2023.
>
> **The hypothesis does not make much sense**
>
> The hypothesis is based on the premise that synthesizing a smaller training dataset from a larger one inherently involves discarding some information and features. We believe this selective reduction is a necessary part of the data condensation process.
>
> **The novelty of the proposed method is not clear.**
>
> GUARD is not just a simple combination of the Dataset Condensation with Gradient Matching (DC) and the CURE method. Unlike the CURE regularizer, our method embeds the regularization into the condensed dataset, allowing for no decrease in actual training time efficiency.
>
> That being said, our work serves as an important foundation for robust dataset condensation and is much more than just the proposed GUARD framework. Not only did we provide a formulation of robust dataset condensation and a comprehensive robustness benchmark of SOTA methods, but we are also the first to theoretically prove that minimizing the adversarial loss of the condensed dataset and minimizing the adversarial loss of the real dataset is only different by a constant factor, which is very useful for future works in this area.
>
> **The Experiment is Weak.**
>
> Our experiments in the main paper focused on MNIST and CIFAR10 because we attached the regularizer to DC, which is notably less effective with high-resolution datasets. However, in the appendix, we extended our experimentation using SRe2L [3], a more suitable condensation method for high-resolution datasets. This allowed us to test our method on ImageNette and present these results for a more comprehensive evaluation
>
> [3] Zeyuan Yin, Eric Xing, and Zhiqiang Shen. Squeeze, Recover and Relabel: Dataset Condensation at ImageNet Scale From A New Perspective. Neurips 2023.
>
> **The improvement is marginal**
>
> We recognize that our improvement may seem low compared to normal adversarial robustness papers. This is because improving adversarial robustness on condensed datasets is not a trivial task. Many of the adversarial defense methods that work well under the real dataset setting do not apply in the condensed dataset setting. This is because one of the fundamental premises of dataset condensation is that no real data will be given to the downstream model at training time, which is important to preserve many of the properties of condensed datasets, including size, training efficiency, and privacy. Without real data, the data distribution that the model learns is already largely different compared to the test images, making it much easier for the adversary to find adversarial examples. Therefore, we don’t think it is realistic to expect the same degree of robustness from condensed datasets and real datasets.
>
> In addition, one reason for the low level of robustness is the inherent low level of clean accuracy. For example, many adversarial defense methods can retain a robustness of around 50%, on a model that can normally achieve 90% clean accuracy on CIFAR10. However, DC can only achieve 50% clean accuracy on CIFAR10, so it is not surprising that when applied the same level of attack, the robustness accuracy deteriorates severely.
>
> In the appendix, we conducted a comparison between our results and those obtained from adversarial training, which is recognized as one of the more effective adversarial defense methods for standard datasets. This comparison demonstrates that our method yields superior results.

---

### Official Review · Reviewer_jErL · 2023-11-02

**Soundness:** 2 fair
**Presentation:** 3 good
**Contribution:** 2 fair
**Rating:** 3
**Confidence:** 4

**Summary:**

The authors study how to compute a condensed dataset on which the trained models achieve good adversarial robustness. Specifically, the authors propose an algorithm GUARD, matching the gradients of the model trained on regularized loss, where the regularizer tries to punish large curvature. Finally the authors compare GUARD with other data condensation methods on MNIST, CIFAR10 and ImageNet under different levels of adversarial attacks.

**Strengths:**

The paper is written clearly and motivation is also clear. I personally think it is worth studying how to compute such condensed dataset which promotes adversarial robustness without adversarial training.

**Weaknesses:**

1. The theoretical part (section 5) is not sound enough.
2. The reference is not comprehensive. The main idea is to match the gradient which tends to have smaller curvature (sharpness). There are many existing literatures on how to reach flatter minima, e.g., sharpness aware minimization, implicit regularization. Though these works are not about condensaton, some of them might have relations to adversarial robustness (like arXiv:2305.05392). Also, there are other works also trying to give such adversarially robust condensation like arXiv:2211.10752, but they are neither mentioned and nor compared.

**Questions:**

1. Is it $\phi$ or $\theta$ in "where $\phi$ is some neural network and D is a distance function in the parameter space" in page 4?
2. The definition of $h$ is missing in Proposition 1.
3. Why is $\tilde \ell(x,v)$ convex in terms of $x$ in the appendix A given that $\ell(x)$ is convex? It's obvious that $\tilde\ell(x,\cdot)$ is convex but $\tilde\ell(\cdot,v)$ might not be convex. Say $\ell(x)=-x^3+3x^2$ and hence $\ell(x)$ is convex in $[0,1]$. The corresponding quadratic approximation $\tilde\ell(x,v)$ has second order derivative on $\frac{\partial^2\tilde\ell(x,v)}{\partial x^2}=-6x$ is non-positive in $[0,1]$, which implies that it is non-convex in $[0,1].$
4. I think eq(12) in the appendix A requires $h^{-1}(\cdot)$ to be a contraction map, why is that true? If it is true, then $\left\|h\left(x^{\prime}\right)-\mathbb{E}_{x \sim D_c}[h(x)]\right\| \leq \sigma$ basically implies that the distance between the $x'$ and the mean of training samples with label $c$ is no larger than $\sigma$. Is it a reasonable assumption?

---

> ### Author Response · Authors · 2023-11-22
>
> We greatly appreciate your constructive feedback. We are especially thankful for your acknowledgment of our motivation and the value of our objectives. Below, we have provided responses to your questions and suggestions:
>
> **The theoretical part (section 5) is not sound enough.**
>
> Regrettably, due to space limitations in the main paper, we had to place the bulk of proof in the appendix. Please view the appendix for a more formal proof of the proposition.
>
> **The reference is not comprehensive.**
>
> Thank you for suggesting papers regarding sharpness aware minimization and other related topics. We agree that it relates to our topic of study, and have included them in our paper.
>
> **Is it $\phi$ or $\theta$ in “where $\phi$ is some neural network and D is a distance function in the parameter space" in page 4?**
>
> Yes, thank you for pointing out our type. It is indeed supposed to be $\theta$.
>
> **Definition of h is missing**
>
> h is a feature extractor, which was defined in the formal proof in our appendix. We have added the definition to the proposition for clarity.
>
> **Why is $\tilde{l}(x, v)$ convex given that l(x) is convex?**
>
> The first term, $\ell(x)$, is convex by assumption. The second term, $\nabla \ell(x)^{\top} v$, is a linear function in $v$ and inherently convex. The third term, $\frac{1}{2} v^{\top} H v$, is a quadratic form. Given that $H$m the Hessian of a convex function $\ell(x)$, is positive semi-definite, the quadratic form is also convex in $v$. As the sum of convex functions retain convexity, the collective convexity of all three terms ensures that $\tilde{\ell}(x+v)$ is convex.
>
> **I think eq(12) requires $h^{-1}()$ to be a contraction map**
>
> The assumption in eq(12) is adopted from a specific dataset condensation method outlined in [1]. This method specifically optimizes the synthetic dataset to minimize the distance between the distribution of the real and synthetic data.
>
> [1] Bo Zhao and Hakan Bilen. Dataset condensation with distribution matching. In IEEE/CVF Winter Conference on Applications of Computer Vision, 2023.

---

> > ### Comment · Reviewer_jErL · 2023-11-22
> > **Need further clarification**
> >
> > Thanks for your response. However, I need more clarifications.
> >
> > 1. "As the sum of convex functions retain convexity, the collective convexity of all three terms ensures that $\tilde \ell(x,v)$
> >  is convex." Yes I agree that $\tilde \ell(x,\cdot)$ is convex. However, eq(8) in Appendix A requires the convexity of $\tilde \ell(\cdot,v)$, which is not generally true. (See the previous counterexample I gave.)
> >
> > 2. Can you specify the paragraph in [1] where the assumption comes from? I cannot find it. Thanks in advance.

---

### Meta-Review · Area_Chair_vUTC · 2023-12-08

**Metareview:**

The submission proposes to condense datasets so that models trained on such data would achieve higher adversarial robustness. However, the reviews raise several questions of the theoretic background, criticize the unclear novelty of the work as well as the limitations in the experiments. The rebuttal did address some concerns on the theoretic background, yet could not address the concerns regarding the limitations in the experimental evaluation.

**Justification For Why Not Higher Score:**

The experimental evaluation of the paper is too weak.

**Justification For Why Not Lower Score:**

N/A

---

### Decision · Program_Chairs · 2024-01-16

Reject